# Biopolymers of Polycaprolactone Loaded with Caffeic Acid and *Trametes versicolor* Extract Induced Proliferation in Human Coronary Artery Endothelial Cells and Inhibited Platelet Activity

**DOI:** 10.3390/ijms26104949

**Published:** 2025-05-21

**Authors:** Diego Fernando Gualtero, Diana Marcela Buitrago, Ana Delia Pinzón-García, Willy Fernando Cely Veloza, Leydy Tatiana Figueroa-Ariza, Santiago Torres-Morales, Juan David Rodriguez-Navarrete, Victor Junior Jimenez, Gloria Inés Lafaurie

**Affiliations:** 1Unidad de Investigación Básica Oral—UIBO, Vicerrectoría de Investigación, Facultad de Odontología, Universidad El Bosque, Bogotá 111321, Colombia; buitragodianam@unbosque.edu.co (D.M.B.); adpinzon@unbosque.edu.co (A.D.P.-G.); leydytatianafigueroa@gmail.com (L.T.F.-A.); 2Facultad de Odontología—UBSIFO, Universidad El Bosque, Bogotá 111321, Colombia; storresm@unbosque.edu.co (S.T.-M.); jdrodriguezn@unbosque.edu.co (J.D.R.-N.); vjimenezpl@unbosque.edu.co (V.J.J.)

**Keywords:** polymer, electrospinning, caffeic acid, *Trametes versicolor*, endothelium, antiplatelet activity, restenosis

## Abstract

In atherosclerosis, the proliferation and migration of endothelial and smooth muscle cells (SMCs) and platelet activation alter endothelial function. Naturally occurring substances, such as caffeic acid (CA) and *Trametes versicolor* extract (TvE), have medicinal properties and are traditionally used for their antiproliferative, antioxidant, and anti-inflammatory effects. Electrospun 5% and 8% polycaprolactone-loaded CA or TvE was developed as a delivery system. Cytocompatibility was evaluated using human coronary artery endothelial cells (HCAECs), coronary artery SMCs (CASMCs), and platelets. Three types of systems (µF-CA, µF-TvE, and µF-CA/TvE) were developed and microscopically characterized. Analysis with scanning electron microscopy showed multidirectional fibers with diameters of 2–4.5 μm. The µF systems were hydrophobic and low cellular adhesion. The viability of CASMCs decreased with microfibers of 8% PCL and high CA concentration. However, the viability of CASMCs and HCAECs improved with 5% PCL and low CA concentration. Treatment with µF-TvE and µF-CA/TvE increased cell viability. HCAEC proliferation was affected by µF-CA, but incorporating TvE improved it. Platelet viability was unaffected by any µF system, but µF-CA and µF-CA/TvE inhibited the activation and adhesion of platelets. The results suggest that microfibers loaded with CA and TvE play a dual role in modifying HCAEC proliferation and blocking human platelet activation and adhesion. These findings have the potential to mitigate the atherosclerotic process.

## 1. Introduction

Poly-ε-caprolactone (PCL) is the most common biopolymer in manufacturing controlled release systems, particularly in producing micronanofibers. PCL is a semicrystalline aliphatic polyester, the most widely utilized synthetic polymer in medical applications. Its primary attributes include a slow degradation rate, biocompatibility, and similarity to natural tissue components, such as collagen and extracellular matrix. PCL has excellent mechanical properties, ease of processing, and compatibility with hard and soft tissues. PCL use has been approved by the FDA. PCL is thermally stable and has excellent mechanical properties [1]. PCL fibers are promising drug delivery systems due to their biocompatibility, biodegradability, and exceptional support matrix characteristics. Many active substances can be encapsulated within these fibers, isolated from the external environment by a physical barrier. Anticancer [2], hypoglycemic [3], and antibiotic [4] substances are among the active substances encapsulated using PCL nanofibers. Upon exposure to biological fluids through enzymatic or nonenzymatic mechanisms, this barrier dissolves and disintegrates, facilitating the release of the encapsulated active agent [5,6]. PCL micronanofibers protect the drug against degradation and extend its circulation time, reduce immunogenicity and toxicity, and enhance drug specificity, selectivity, etc. [7].

Caffeic acid (CA) is a phenolic compound in coffee, potato, wine, tea, and other natural remedies, such as propolis. CA and its derivatives exhibited antioxidant, anti-inflammatory, and anticancer activities and exerted antimicrobial and cytostatic effects [8,9]. CA antagonized endothelin-1 and was reported as a promising antihypertensive drug [10]. CA was indicated to benefit cholesterol balance, elevating serum HDL-C. CA exhibited a local anti-inflammatory effect, reducing the levels of serum proinflammatory cytokines, such as TNF-α, IL-6, and MCP-1 [11]. CA reduced thrombus formation in mouse cerebral arterioles in vivo and inhibited thromboxane A_2_ production [12], αIIbβ3 integrin activation, mitogen activation, activated protein kinase phosphorylation, and granule secretion; increased cyclic adenosine monophosphate levels in rat or human platelets in vitro [13]; and inhibited cyclooxygenase activity and P-selectin adhesion expression in platelets in vitro. CA inhibited collagen-induced platelet aggregation through the phosphorylation of the AMP-dependent inositol 1,4,5-trisphosphate receptor and inhibited platelet-mediated clot retraction [14]. These findings suggest that CA is an excellent starting point for developing therapeutic agents against thrombotic disorders.

The extracts of the fungus *Trametes versicolor* are used in traditional Chinese medicine for cancer treatment [15,16]. The active components in the extract are polysaccharide peptides, krestin polysaccharides, phenolic compounds, and flavonoids. The antioxidant effects of *T. versicolor* are attributed to the presence of phenolic compounds and flavonoids. By contrast, polysaccharide peptides and krestin polysaccharides are associated with cytotoxic, antiproliferative, and anti-inflammatory effects. *T. versicolor* extracts exhibited cytotoxic and anti-inflammatory effects in breast cancer and melanoma cells [17,18]. MCF-7 cells treated with *T. versicolor* extract showed increased lactate dehydrogenase (LDH) release and reactive oxygen species production. Data on the cardiovascular effects of *T. versicolor* are lacking, although several medicinal mushrooms are known to act as antihypertensives and antiplatelet agents and are cardioprotective [19]. For the first time, Nikolic et al. demonstrated the cardioprotective effects of *T. versicolor* heteropolysaccharides both in vivo and ex vivo, mediated through blood pressure and oxidative stress lowering effects and beneficial effects on cardiac function, together with hypoglycemic properties [20], which is a starting point for further evaluation of the cardioprotective effects and roles of *T. versicolor.*

Ischemic cardiovascular disease and myocardial infarction are the leading causes of death worldwide in >50-year-old individuals, according to the 2019 Global Burden of Disease Report [21]. A recent longitudinal study on atherosclerotic disease from 1990–2019 concluded that the global incidence of peripheral arterial disease, a primary clinical manifestation of atherosclerosis, has increased in >50-year-old individuals [22]. Atherosclerotic endothelial dysfunction is associated with inflammation and oxidative stress caused by vascular tissue injuries induced by various factors, including oral infections [23,24]. During advanced atherosclerosis, endothelial cells lose their regenerative capacity, SMCs proliferate and migrate to the intima, and platelets undergo activation and aggregation, leading to plaque rupture and subsequently thrombosis [25,26].

The use of drug delivery systems based on polymers, such as PCL, whose primary function is to support active substances of interest, can improve both the stability and control of drug release, increasing their therapeutic potential by reducing their degradation before reaching target tissues [27]. A system developed using PCL electrospun to support CA and *T. versicolor* extract could mitigate vascular problems related to activation and aggregation platelets, like intra-stent restenosis [28]. Incorporating natural components, such as a *T. versicolor* extract or CA, previously studied in traditional medicine for other chronic diseases (e.g., cancer), in drug delivery by scaffold PCL is an essential way in which biotechnology can be applied in healthcare. Therefore, bioresorbable devices covering drug-eluting stents has potential application in the treatment of intrastent restenosis [29]. Therefore, searching for agents that attenuate cellular changes in vascular tissues could be a therapeutic target for treating atherosclerosis. In this study, PCL microfibers were prepared by electrospinning, loaded with CA, and coated with a chitosan solution containing *T. versicolor* extract to evaluate their cytocompatibility in human coronary artery endothelial cells (HCAECs), human coronary artery SMCs (CASMCs), and human platelets. In addition, antiplatelet activity was evaluated.

## 2. Results

### 2.1. Morphological Characterization

The fibers were morphologically characterized and geometrically evaluated (size distribution by diameter and orientation) using scanning electron microscopy. Factors, such as viscosity, polymer concentration, type of solvent, and operational factors (e.g., distance between the needle and collecting plate, applied current, and leakage flow of polymer solution) influenced fiber morphology and size.

Figure 1 shows the scanning electron microscopy images of PCL fibers with a magnitude of 400× and a scale of 100 μm (a), a magnitude of 5000× and a scale of 10 μm (b), and a magnitude of 20,000× and a scale of 2 μm (c). Table 1 shows the size distribution for each microfiber type evaluated. The morphological characteristics and size distribution of different PCL microfibers alone or loaded with CA exhibited random directions. This effect resembles electrospun fibers produced with a static rectangular metal collector.

The micrographs show the effect of the polymer, drug, flow rate, and voltage on the morphology of PCL fibers. The µF fibers had a size distribution between 480 nm and 4.60 µm and an average diameter of 2.50 µm with uniform appearance. In comparison, the uF-CAF-1 fibers had an average diameter of 1.96 µm, with uniform morphology and no imperfections. This increase in the size of the µF nanofibers can be explained by the injection rate of the polymeric mixture used (7 mL/h) and the high concentration of the PCL polymeric solution as different authors report that high injection rates (>0.5 mL/h) and high concentration (>8%) result in formation of larger droplets at the tip of the needle, producing fibers with a larger diameter (Zong et al., 2020) [30].

### 2.2. Determination of Contact Angle

Figure 2 presents images of contact angle measurement. Surfaces with contact angles of >90° and <90° were classified hydrophobic and hydrophilic, respectively, using water as solvent.

The contact angles of the surfaces of the µF and y µFCAfibers were measured to assess their wettability and hydrophilicity. PCL microfibers with or without CA presented a contact angle >90° or ≥90° (Table 1). The contact angle of the fibers could be determined only without the chitosan coating as the hydrophilic nature of the chitosan film did not allow droplet formation on coated fibers (µF-TvE and µF-CA-1/TvE).

### 2.3. Quantification of Total Polyphenol and Protein Delivery from Coated Fibers

First, calibration curves were established for each method. Caffeic acid was used as a reference in the Folin–Ciocalteu assay, while *Trametes versicolor* extract was used in the Bradford assay (Figure 3).

Data obtained from the Folin–Ciocalteu assay indicated that the microfiber samples without *Trametes versicolor* extract (µF-CA-4, µF-CA-1) exhibited diffusion-controlled behavior, with release percentages of 81.3% and 98.7%, respectively, as shown in Figure 3C. Additionally, µF-CA-4 and µF-CA-1 exhibited a release constant kH of 1.38 mg/h^1^/^2^ and 1.39 mg/h^1^/^2^, respectively.

In contrast, no diffusion-controlled release of total polyphenols was observed for the samples loaded with *Trametes versicolor* extract (µF-CA-4/TvE, µF-CA-1/TvE) (Figure 3C). This behavior is attributed to the chitosan-TvE layer, which forms a barrier that limits the diffusion of caffeic acid. Moreover, µF-CA-1 exhibited 86% higher polyphenol release than µF-CA-4 at 24 h and 76.7% higher at 48 h. Similarly, µF-CA-1/TvE showed a 66% higher polyphenol release than µF-CA-4/TvE at 24 h and 75% higher at 48 h. Due to the lower polyphenol dosage in µF-CA-4 and µF-CA-4/TvE, Section 2.4 reports “higher cell viability in CASMCs”.

For total proteins quantified using the Bradford assay, the Korsmeyer–Peppas model showed a relative linearity ranging from 63% to 92% (Figure 3D). Similar to the polyphenol-loaded samples, protein-loaded samples exhibited a higher release rate at 24 h. However, at 48 h, a decrease in protein concentration in the medium was observed, revealing a complex interaction between the analytes and the matrix. This reduction in concentration, up to 61.2%, aligns with the cell viability results reported in the following section at 48 h of analysis.

### 2.4. Cytotoxic Effects of µF-TvE and/or µF-CA on CASMC and HCAEC Cultures

Assessing the safety of microfibers loaded with *T. versicolor* and CA is a crucial aspect of their characterization. The potential toxic effects of empty microfibers or those loaded with TvE and CA were evaluated using an appropriate in vitro cell model to study endothelial and muscle functions in various physiological and pathological processes of cardiovascular and cerebrovascular diseases.

The cytotoxicity results demonstrated that empty 8% PCL microfibers (91.3%) did not affect the viability of CAMSCs at 24 h of treatment. However, at 48 h, a cytotoxic effect of 61.2% was observed (Figure 4), which was not statistically significant compared with the control group (86%). Regarding 8% PCL microfibers loaded with CA and *T. versicolor*, after 24 h of treatment of CASMCs, a cytotoxic effect was present with µF-CA-1 (28.4%) compared with µF-TvE (46.8%) and µF-CA-1/TvE, which exhibited 47.5% cell viability compared with the cells from the control group (without treatment, 91%). At 48 h, similar findings were observed where different treatments induced a cytotoxic effect. The most significant effect was observed with the µF-CA-1/TvE (21.6%), followed by µF-CA-1 (41%), and µF-TvE (56.2%), compared with 86% in the control group (*p* < 0.05) (Figure 4B). The results of the resazurin assay were validated in vitro (Figure 4C,D). The integrity of the cell membrane was altered, as evidenced by the release of LDH. At 24 h, cells treated with µF-CA-1 and µF-CA-1/TvE exhibited a significant release of LDH (36.0% and 19.2%, respectively), compared with the control group (0.0%) (*p* < 0.05). Similarly, at 48 h, the same treatments increased LDH release by 93.5% and 64.5% in CASMCs stimulated with µF-CA-1 and µF-CA-1/TvE, respectively (*p* < 0.05). Thus, these systems exerted a cytotoxic effect on CASMCs.

Changes were implemented to the delivery system, and cell viability after microfiber treatment was improved. Under the newly established conditions, µF was developed with 5% PCL and CA to the contractions of CA-4. After 24 h of incubation, the viability of CAMSCs declined; however, this effect was not statistically significant for cells treated with µF-CA-4 (67.2%) and µF-TvE (51.2%) compared with untreated cells (86.2%). By contrast, cells treated with µF-CA-1/TvE exhibited an increase in cellular metabolism (106.2%) (Figure 5A). After 48 h of treatment, no treatment affected cell viability (Figure 5B). The results of viability analysis correlated with LDH release data (Figure 5C,D). Compared to the control, µF (22%) affected the culture at 24 h (*p* < 0.05). There was some release of LDH with µF-CA-4 (9.5%) and µF-TvE (5.0%), but this effect was not statistically significant. After 48 h, LDH release was insignificant compared with untreated cells. These results suggest that the most critical release of CA and TvE occurred during the first 24 h of treatment, leading to cell recovery. Nevertheless, further studies are necessary.

We evaluated the effects of microfibers under the conditions of 5% PCL and CA-4 in HCAECs because it exhibited the best safety conditions in CASMCs (Figure 6). The data demonstrated that cell viability was affected only by µF-Ca-4/TvE, with a decrease in viability of 60% compared with the control group (98.3%) (Figure 6A, *p* < 0.05). However, at 48 h, there was a significant increase in cell proliferation with µF-TvE (166%) compared with untreated cells (119%) (*p* < 0.05). In particular, empty µF (31%) (*p* < 0.05) induced cytotoxic effects after 48 h of incubation (Figure 6B). Regarding the results of LDH release in the presence of microfibers, no significant changes were observed in any of the systems evaluated at 24 and 48 h, with the percentages of LDH of <1% (Figure 6C,D). These results are consistent with the viability data described. However, in the case of µF, a decrease in viability is observed over a 48-h period without an increase in LDH. These results suggest that this polymer can affect cellular metabolism without inducing cell death.

### 2.5. Cell Adhesion of HCAECs to Microfibers

A cell adhesion assay was conducted to determine whether the delivery systems allowed the adherence and seeding of HCAECs. PECAM-1 was used as an endothelial marker and nuclei were detected with DAPI staining through an immunofluorescence assay (Figure 7). After 7 d of culture, no adhesion of HCAECs to µF, µF-CA-1, and µF-CA-4 was observed (Figure 7B–D). Cell adhesion was observed in the culture without the μF system (Figure 7A). Based on these results, the delivery systems μF without and with CA do not allow the adherence of HCAECs.

### 2.6. Effect of Microfibers on HCAEC Proliferation

The effects of the μF delivery systems on the proliferation of HCAEC cultures is shown in Figure 8. After 24 h of treatment, the μF-AC system increased cellular proliferation compared with that in unstimulated cells (*p* < 0.05). By contrast, the μF-CA/TvE system decreased proliferation, similar to the control with everolimus (Figure 8A). After 48 h of treatment, the HCAEC culture treated with μF-CA maintained proliferation, but it was increased with μF-CA/TvE compared with the control (*p* < 0.05). In Figure 8B, LPS-A.a was used as the proliferation control.

### 2.7. Effect of Microfibers on Platelet Viability

The viability of platelets was evaluated using fluorescence microscopy, employing the LIVE/DEAD assay. None of the microfibers affected platelet viability, with the majority of green platelets observed, similar to the control group (Figure 9A). Regarding morphological characteristics, the platelets in the control group were oval, and no platelet aggregates are observed. By contrast, the group of platelets stimulated with *P. gingivalis* LPS (Figure 9B), empty µF (Figure 9C), and loaded with µF-CA-4/TvE (Figure 9F) demonstrated platelet accumulation or aggregates. Thus, *P. gingivalis* LPS induced platelet aggregation and µF of 5% PCL, both empty and loaded with µF-CA-4/TvE, probably could not inhibit this aggregation. Interestingly, µF-CA (Figure 9D) and µF-TvE (Figure 9E) inhibited platelet aggregation induced by *P. gingivalis* LPS. However, further studies are required to microscopically confirm this observation and investigate the effects of inhibiting platelet aggregation. The results demonstrated that empty or loaded µF with different treatments did not affect platelet viability when compared with unstimulated platelets (Figure 9G), indicating the safety of these prototypes on the platelets.

### 2.8. Inhibition of Platelet Activation Treated with Microfibers

The effects of the microfiber systems on platelet activation were evaluated using PAC-1 (a monoclonal antibody that detects activation through the αIIbβ3 pathway) and the CD40 receptor (CD40-R) stimulated with *P. gingivalis* W83 LPS. The data revealed that μF-CA-4 microfibers had inhibitory effects on the expression of double-positive PAC-1/CD40 (6.30%) as well as the μF-CA-4/TvE system (3.4%) compared with the negative control (untreated platelets, 1.87%) (*p* < 0.05) (Figure 10); however, the effects were not greater than those presented by pure CA molecules (1.85%). In addition, this compound could independently inhibit PAC-1 and CD40 (*p* < 0.05). By contrast, the different microfiber prototypes did not modify the activation of PAC-1 and CD40 induced by *P. gingivalis* LPS (Table 2).

### 2.9. Effect of Microfibers on Platelet Adhesion

The effect of microfibers loaded with CA, TvE, or µF-CA/TvE on platelet adhesion molecules, P/E-selectin, and PECAM-1 was evaluated in human platelets stimulated with LPS from *P. gingivalis* W83 (Figure 11, Table 3). The data revealed that CA, CA-, and TvE-microfibers inhibited the expression of PECAM-1 (*p* < 0.05). In comparison, the expression of P/E-selectin was inhibited only by μF-CA (*p* < 0.05).

## 3. Discussion

Atherosclerosis is a disease of vascular tissue in which endothelial cells, SMCs, platelets, and inflammation play relevant roles in the progression of disease [31]. The proliferation and migration of SMCs to the artery lesion promote the growth of atherosclerotic plaques, and their rupture induces platelet activation and aggregation, leading to thrombosis and stroke. Therefore, it is necessary to develop strategies that inhibit SMC proliferation and platelet activation and improve vascular endothelium to mitigate this process. Natural substances, such as CA and TvE, have potential antioxidant, antiproliferative, anti-inflammatory, and tissue regeneration activities [32,33,34,35]. This study evaluated the potential use of CA and TvE in PCL fibers, like delivery systems, in vascular cells.

Three prototypes of microfibers loading CA and TvE (µF-CA, µF-TvE, and µF-CA/TvE) were developed using electrospun PCL. Morphological analysis revealed that the microfibers had diameters of 2.0–2.5 µm, with multidirectional fibers, and those have not been altered by loading with CA or TvE. The hydrophobicity of PCL microfibers did not change with CA, as was observed with contact angle measurement, and it was confirmed with the low cellular adhesion of HCAECs with different prototypes of µF. By contrast, VASMCs cultured on scaffold PLC with varying sizes of fiber (0.5–10 µm) by Han et al. (2019) exhibited adhesion, viability, and proliferation [36].

Furthermore, the decrease in the diameter of the µF-CAF fibers can be attributed to the presence of a molecule with polar characteristics, such as CA, and the type of solvent used in the preparation of the polymeric mixture (methanol, chloroform, and dichloromethane), as this mixture has high relative permittivity and high polarity, which is a determining factor in fiber formation despite constant flow and voltage conditions [30,37]. At the same time, the microphotograph shows the formation of some defects (3a–c) that can be explained by both the polymer injection rate (7.0 mL/h) and the volume of the natural polymer chitosan solution used in the mixture and the coating film [38]. This typically occurs when using natural polymers in nano and microfiber preparation [38]. Another factor that acts as a determinant of the formation of nanofibers is viscosity; high viscosity prevents the formation of smooth and uniform fibers.

The presence of chitosan (in low concentrations) did not allow the proper formation of coating fibers. Other authors have reported similar findings, identifying problems in fiber formation in PCL with polymers, such as alginate and chitosan, as these polymers have positive and negative charges that produce fibers with defects and agglomerates. Furthermore, in this case, the chitosan solution was used as an external coating, which can explain the low fiber formation capacity of this biopolymer, which was rigid in an aqueous solution [39].

PCL is hydrophobic, possibly owing to the hydrophilic carbonyl groups in its structure. The contact angle of the µF-CA-1 fiber was smaller than that of the µF fiber, with a hydrophobic nature [40,41], likely due to the highly hydrophilic COOH moiety of CA on the surface of the former [42]. Nanofiber wettability may improve cell proliferation and biocompatibility [43]. In a similar study, the incorporation of natural acids and derivatives as well as other drugs into PCL nanofibers also increased nanofiber hydrophilicity and provided an excellent release profile [44].

The release profile of total polyphenols was evaluated using the Higuchi and Korsmeyer-Peppas kinetic models. Higuchi’s models describe the diffusion of bioactive compounds from the polymer matrix. While the total protein release model described by the Korsmeyer–Peppas model reveals diffusion and probable disintegration or erosion of the chitosan coating, since it is a superficial layer of the microfiber. The comparative analysis of different formulations (µF-CA-1, µF-CA-1/TvE, µF-CA-4, µF-CA-4/TvE) revealed variations in release rates, where higher polyphenol concentrations led to greater release efficiency. Furthermore, the presence of *Trametes versicolor* extract influenced the release profile, suggesting interactions between the bioactive compounds and the polycaprolactone matrix.

In this study, 8% and 5% PCL affected cell viability and CASMC cytotoxicity was enhanced by µF-CA-1 and µF-CA-1/TvE but not by µF-TvE. The aforementioned parameters exhibited an improvement when the CA concentration was µF-CA-4. The cytotoxic effect of PCL may be associated with the proportion of the polymer that generates acids and other byproducts, which at high concentrations could contribute to an acidic environment that negatively affects cell viability [45,46]. Moreover, the accumulation of degradation products may impact the mechanical and biological properties of the scaffold, thereby affecting its capacity for cell regeneration. Conversely, PCL has been demonstrated to induce LDH release without inducing cellular necrosis; previous studies have demonstrated that PLC can influence mitochondrial activity or cellular metabolic activation by affecting ATP production and generating oxidative stress, without compromising the integrity of the plasma membrane [47], which may be exacerbated in conditions in which PCL microfibers are modified or loaded with bioactive compounds, thus increasing cytotoxicity [42,48]. This may be associated with the effects seen in µF-TvE. Similarly, the variability in the cellular response to PCL in previous studies may be influenced by the cell type utilized [48,49]. Different cell lines may exhibit diverging tolerance thresholds to cytotoxicity, indicating that HCAECs and CASMCs may exhibit low tolerance to PCL.

At high concentrations, CA exerts cytotoxic effects on endothelial and muscle cells and can influence cell proliferation and apoptosis [50]. These effects are likely to impact metabolic oxidation [51]. Prior research has demonstrated that PCL nanofibers encapsulating CA increase HCAEC viability compared with free CA [52]. This has also been documented by Bellosta et al. (2022) [53]; the present results show unaltered cell viability, suggesting that controlled delivery through nanofibers can be advantageous. In contrast, Kim et al. (2020) described how CA-loaded PCL nanofibers affect smooth muscle cell proliferation and viability, a result comparable to ours where CA-1 induced cell death, and a low CA-4 concentration altered this effect. Those studies suggest that controlled release of CA can enhance biocompatibility [54].

Some studies have indicated that *T. versicolor* extracts may enhance cell viability in human cell lines [55], including endothelial and smooth muscle cells, by mitigating the detrimental effects of free radicals. The findings of this research indicate that TvE induces moderate metabolic changes in CASMC and HAEC cells at 24 h of treatment, without a significant increase in LDH release. This suggests that its cytotoxic effect is not associated with cellular necrosis, but rather with possible mechanisms such as metabolic stress, inhibition of proliferation, or early apoptosis. The bioactive metabolites of *T. versicolor*, including polysaccharides and phenolic compounds, have been documented to modulate cellular activity by inducing oxidative stress, cell cycle alterations, and the activation of apoptotic pathways [56]. Furthermore, *T. versicolor* extracts have demonstrated antiproliferative effects in tumors and normal cells by regulating key cell cycle proteins, such as p53 and Bcl-2, suggesting that the cellular response depends on factors such as extract concentration and duration of exposure, without causing immediate damage to the cell membrane [57]. The combination of these factors suggests that µF-TvE may be interfering with mitochondrial function and cell cycle progression in CASMCs, which would explain the reduction in cell viability without marked LDH release or evident necrogenic effects. In order to confirm this mechanism, further studies are required that evaluate the expression of apoptotic biomarkers, analyses cell cycle progression, and measure mitochondrial potential.

Concerning cellular proliferation, µF-CA-4 and µF-CA-4/TvE promoted HCAEC proliferation. Treatment of HUVECs with 1 µM and 100 nM CA induced the release of nitric oxide and inhibited the production of reactive oxygen species [32]. By contrast, although TvE treatment did not inhibit the production of reactive oxygen species, it had anti-inflammatory effects and inhibited cellular migration in HUVECs [17]. Although we did not evaluate proliferation in CASMCs with µF-CA/TvE, human CASMCs treated with CA inhibited cellular proliferation and migration and induced apoptosis, blocking AKT-1, MEK-1, and ERK-1 signaling [58]. These results suggest that microfibers loaded with CA/TvE can reduce the dysfunction of endothelial migration and inflammation associated with atherosclerosis.

Substances that can have antithrombotic effects have significant therapeutic potential. CA exhibited antiplatelet effects, reduced thrombus formation, and inhibited the production of thromboxane A_2_, activation of IIb, production of P-selectin, and secretion of platelet granules [12]. However, its bioavailability, absorption, and release are very low, which makes the development of a release system that allows for improvements in these factors of great therapeutic interest. Studies have shown that CA is noncytotoxic for platelets. Although it is not easy to evaluate platelet viability because they are anucleated, studies assessing the release of LDH or mitochondrial respiration have shown that CA and its derivatives do not induce cytotoxic effects in human platelets [14], which is consistent with our results using the LIVE/DEAD assay, where CA and different microparticle systems did not induce toxic effects. Concerning the system loaded with *T. versicolor*, there are no data reported in the literature that can be compared with this study. This study used LPS-Pg as an inducer of platelet adhesion and activation. *P. gingivalis* LPS produces various inflammatory and immune mediators that can induce platelet activation in two ways: directly through the release of DC40L or indirectly through the release of endothelial tissue factor, which activates αIIbβ3 in platelets [59,60].

Pathophysiological platelet adhesion and activation in the vasculature can lead to coagulation and, ultimately, thrombosis. Antiplatelet drugs are used to prevent and treat thromboarterial diseases. Phenolic compounds, such as CA and ferulic acid, are known antiplatelet agents; however, this clinical effect is rapid because they are intensely metabolized and degraded into metabolites with low activity. Thus, a PCL microparticle system could provide controlled release and improved biological impact. The results of this study showed that the system of PCL microparticles loaded with CA inhibited platelet adhesion and aggregation (specifically of PECAM-1), release of CD40, and inhibition of integrin αIIbβ3, as reported by Nam GS et al. (2018, 2020) [12,14].

Regarding the development of release systems with CA, no studies have been carried out on the effects of platelet aggregation. Still, systems of hybrid nanofibers of polyester, urea, and silk fibrin incorporating ferulic acid as a compound of interest have been studied [61]. The results showed that this vascular graft had an antioxidant effect and blood compatibility and inhibited platelet adhesion in the system, comparable to the results proposed in this study. The effects of the phenolic compounds were mainly related to the inhibition of platelet activation induced by arachidonic acid and the inhibition of thrombin production. Regarding cardiovascular effects and platelet aggregation, we do not have enough data to compare the results described in this study of *T. versicolor* microfibers and CA/TvE. Still, our results are interesting because the TvE system inhibited platelet adhesion by acting on PECAM-1 and P/E-selectin. We hypothesize that this result is related to the antioxidant effect of TvE [62], where an antioxidant can reduce platelet activity by eliminating lipid peroxides and radicals that damage the arterial endothelium, inhibiting prostacyclin synthesis or specific interactions [63].

Therefore, antioxidants could theoretically have antiplatelet and anticoagulant activity, and new studies are needed to validate this theory. The effect shown by the system containing CA/TvE is of great interest. Both compounds could inhibit platelet adhesion and activation molecules and may be associated with a synergistic mechanism. This requires the continued evaluation of these systems with cardiovascular therapeutic potential.

## 4. Materials and Methods

### 4.1. Preparation of Solutions for Producing PCL Fibers and the Process of Electrospinning

PCL (Mw = 80,000 g/mol), CA), and high molecular weight chitosan were purchased from Sigma-Aldrich (Sigma-Aldrich, St. Louis, MO, USA). The solvents dichloromethane, methanol, and chloroform were purchased from Merck (Rahway, NJ, USA). All other materials used were of analytical grade. The extract of *T. versicolor* was obtained from Green Health Center (Yanta District, Xi’an, China, www.greena-bio.com, accessed 30 October 2020, INVIMA registration: NSA-001154-2016). Dry material with an average molecular weight of 10,000–200,000 was obtained from the fruiting body and mycelium. The chemical components present in the extract include 50% polysaccharides, 10% beta D-glucan, and 40% polysaccharides/protein, as specified by the manufacturer.

The fibers were prepared with 500 or 800 of PCL dissolved in 10 mL of a mixture of solvents to generate a base polymer solution. The solutions were loaded with different concentrations of CA (CA-1 and CA-4), stirred for 3 h at room temperature, and electrospun. Electrospinning was performed with a solution flow rate of 7.0 mL/h and a voltage of 10 kV. The fibers were collected on a collector wrapped in aluminum foil and kept at a distance of 15 cm from the needle tip (NanoFiber 100). For *T. versicolor*, fibers were incorporated as a cover film. Then, a 1% chitosan solution in acetic acid and 2.5% *w*/*v* of *T. versicolor* was prepared. PCL fibers, with and without the active agent CA, were coated by immersion in 1 mL of this solution for 2 min and dried at 25 °C for 5 h. Finally, the fibers were stored for physicochemical characterization and cytotoxicity tests.

### 4.2. Physicochemical Characterization of Fibers

#### 4.2.1. Scanning Electron Microscopy

The average diameter and morphology of the electrospun fibers were determined with an FEG-Quanta 200 FEI scanning electron microscope at an acceleration voltage of 20 kV. The fiber samples were coated with a 5-nm thick layer of gold using a spray coater. Average fiber diameters were obtained with ImageJ software version 1.54k using at least 50 measurements.

#### 4.2.2. Contact Angle Measurement

A contact angle measurement system was used to evaluate the wettability (hydrophilicity or hydrophobicity) of the electrospun fibers. The microfibers were placed in a sample holder, and 10 μL water was dropped on their surface using a pipette. A camera recorded the image of the droplet, and the average contact angle of three measurements was obtained using ImageJ software, specifically the Low Shape Axisymmetric Drop Shape tool in ImageJ.

### 4.3. Culture of HCAECs and CASMCs

HCAECs and CASMCs were purchased commercially from Lonza (Walkersville, MD, USA). HCAECs were cultured in endothelial cell basal growth medium (EBM-2; Lonza) supplemented with 5% fetal bovine serum and vascular endothelial growth factors (EGM-2 MV BulletKit, CC-3202; Lonza). CASMCs were cultured in smooth muscle growth medium (SMGS; Lonza) supplemented with 5% fetal bovine serum and muscle growth factors (Abs sample—Control low) (BulletKit, Lonza) under controlled conditions of humidity and temperature (5% CO_2_ and 37 °C). Cells were cultured to 80% confluence, detached using 0.25% trypsin and 0.5 mM EDTA in TBS, seeded at a density of 10,000 cells per well in 96-well plates, and incubated for 20 h to establish stable culture conditions before commencing treatments.

### 4.4. Stimulation of HCAECs and CASMCs with Microfibers Loaded with TvE and CA

HCAECs and CASMCs cultured on 96-well plates were stimulated with microfibers loaded with TvE (µF-TvE), CA (µF-CA), or a combination of both (µF-CA/TvE). Incubation periods of 24 and 48 h were employed. Negative controls consisted of unstimulated cells and cells exposed to empty fibers (µF). Triton-X at 1% served as a control for cell death.

### 4.5. Cell Viability Assay

The alamarBlue assay (Biosource, Camarillo, CA, USA), in which living cells metabolize resazurin and convert it to resorufin [64], was used to assess the viability of HCAEC and CASMC cultures. Following stimulation, microfibers were extracted from each well using previously sterilized non-gripping forceps and culture medium. After prototype removal, a 1-min washing step with PBS was performed. Next, 100 µL of 44 µM resazurin solution in unsupplemented medium (EBM-2 or SMG) was added, and the mixture was incubated at 37 °C for 4 h. Fluorescence readings were recorded at 570 nm with a 630 nm differential filter utilizing a plate reader (TECAN Infinite 200 PRO, Männedorf, Switzerland). Data were converted into viability percentages and analyzed using GraphPad Prism 6 statistical software (GraphPad Corp., San Diego, CA, USA). Each experiment was performed as three independent tests.

### 4.6. Lactate Dehydrogenase (LDH) Activity Assay

Cell cytotoxicity was assessed by measuring LDH release in the cell supernatant after stimulation using the LDH-cytotoxicity Assay kit (Novus, NBP2-54851, Centennial, CO, USA). Supernatants were collected in 1.5-mL Eppendorf tubes and stored at −20 °C for preservation. To each 10 µL of supernatant, 100 μL of the reaction mixture (95.6 μL diaforase/NAD+ mixture catalyst and 4.4 μL iodotetrazolium chloride and sodium lactate staining solution) were added. The resulting mixtures were incubated for 30 min at room temperature in agitation under darkness. Fluorescence was measured at 490 nm using a plate reader (TECAN Infinite 200 PRO, Switzerland). Three independent assays were performed in duplicate. Cytotoxicity percentage was calculated using the following formula:$$Cytotoxicity\;(\%)=\frac{\left(Abs\;sample-Control\;low\right)}{\;(Abs\;Control\;High-Control\;low)\;}\times 100$$

### 4.7. Adhesion of Endothelial Cells to CA-Loaded PCL Systems

A cell adhesion experiment was performed through an immunofluorescence assay to investigate whether the release systems facilitated the adhesion of HCAECs. Fibers loaded with two concentrations of CA (CA-1 and CA-4) were manufactured through electrospinning onto glass coverslips, dried, and sterilized by UV radiation for 15 min. These coverslips were placed in 12-well plates, seeded with 10,000 HCAECs per coverslip, and left to adhere for 2 h. Subsequently, HCAECs were cultured for 7 d at 37 °C and 5% CO_2_ in EBM supplemented with specific growth factors, with medium changes every 48 h. In parallel, HCAECs were seeded on coverslips without the release system as control. Following the culture period, coverslips were fixed using 4% paraformaldehyde, washed twice with PBS, and incubated with FITC antibodies targeting PECAM-1 (1/250 dilution IgG1, Santa Cruz Biotechnology, Inc., Dallas, TX, USA) for 16 h at 4 °C in a blocking solution. After two additional PBS washes, the coverslips were incubated with DAPI for nuclear staining. Sample analysis was conducted using fluorescence microscopy (Zeiss, Imager.M2, Hamburg, Germany).

### 4.8. Cell Proliferation Assay

HCAECs were seeded in 96-well plates at a density of 4000 cells per well. After 16 h of adhesion, the medium was replaced, and the cells were treated as follows: unstimulated cells, Lipopolysaccharide from *Aggregatibacter actinomycetemcomitans* (LPS-A.a; 1.0 µg/mL), everolimus (74 nM), LPS-A.a + everolimus, µF, µF-CA (µF-CA), µF-TvE, µF-CA/TvE, and Triton-X-100 (0.01%). Treatment durations included 24, 48, and 72 h. The experiment was conducted in quadruplicate in two independent experiments (*n* = 8). After the treatment was complete, the cultures were washed twice with PBS (pH 7.4) and mixed with 100 µL of CellTiter 96 AQueous One Solution Cell Proliferation Assay solution per well (Promega, Madison, WI, USA). Incubation was carried out for 4 h at 37 °C and 5% CO_2_, and absorbance was read at 490/630 nm (TECAN Infinite Pro M200). The results are presented as absorbance units, and graphs were created using GraphPad Prism 6.

### 4.9. Platelet Stimulation Model Using LPS from Porphyromonas Gingivalis W83

Human platelets were obtained from platelet-rich plasma from healthy volunteers with informed consent, acquired from Fundación Hematológica Colombia through an inter-institutional agreement between Universidad El Bosque and Fundación Hematológica Colombia. In total, 200 µL of suspended platelets (250,000–350,000 cells/µL) were stimulated with LPS from *P. gingivalis* W83 (3.5 µg/mL) for 1 h and treated with various microfiber types.

### 4.10. Evaluation of Platelet Viability

The effects of *P. gingivalis* W83 and the systems on platelet viability were evaluated using a kit measuring the enzymatic conversion of calcein (LIVE/DEAD Viability/Cytotoxicity Kit; Invitrogen, Carlsbad, CA, USA), according to the manufacturer’s specifications. The platelet suspension was incubated for 1 h at room temperature with 2.5 µM calcein in PBS. Fluorescence was read on a TECAN (Infinite 200 PRO), using excitation and emission filters of 494 and 517 nm, respectively. In addition, platelet viability was evaluated using fluorescence microscopy (Zeiss, Imager.M2, Hamburg, Germany), where live cells produced green fluorescence and dead cells produced bright red fluorescence. Three independent assays were performed in triplicate. Triton-X100 at 1% was used as a death control, and unstimulated platelets were used as a viability control.

### 4.11. Determination of the Expression of Platelet Adhesion and Activation Molecules

Markers for the activation and adhesion of platelets stimulated with different treatments were evaluated. Platelets were suspended in 100 μL of PBS and incubated for 15 min at room temperature with an anti-CD26P monoclonal antibody. (P/E-selectin, Cell Signaling, Danvers, MA, USA), anti-PAC-1, CD40, and PECAM-1, labeled with FITC and PE (Cell Signaling, Danvers, MA, USA). Subsequently, the samples were analyzed by flow cytometry (BD Accuri C6) according to the parameters of size, complexity, and fluorescence intensity for each marker evaluated. Unlabeled platelets were utilized to control the technique and normalize the results internally. Three independent assays were conducted in duplicate.

### 4.12. Statistical Analysis

The results are presented as mean ± standard deviation. Data were analyzed using two-way ANOVA (times and treatments) to compare selected pairs of groups, followed by multiple comparison tests. Non-normally distributed data were analyzed using Dunnett’s nonparametric test and Tukey multiple comparisons. Statistical analysis was performed using GraphPad Prism 6 software. A significance level of *p* < 0.05 was statistically considered.

## 5. Conclusions

This research project was concerned with developing electrospun polycaprolactone (PCL) microfibers loaded with caffeic acid and coated with chitosan solution and *T. versicolor* extract. The objective was to ascertain whether this strategy is an effective means of mitigating the progression of atherosclerosis. The results demonstrated that these microfibers, as evidenced by morphological analysis, maintained adequate characteristics even with the inclusion of bioactive compounds. The results demonstrated that these microfibers maintain suitable characteristics, as evidenced by the morphological analysis, even with the incorporation of bioactive compounds. In this context, thorough characterization of the materials was essential to ensure the fibers’ stability, integrity, and functionality. The pristine PCL microfibers and the chitosan- and TvE-loaded variants were characterized using advanced techniques, such as scanning electron microscopy (SEM) for morphological analysis and contact angle. These analyses provided insights into the surface properties and the uniform distribution of the bioactive compounds within the fibers.

Regarding biocompatibility, CA-microfibers and TvE-microfibers exhibited contrasting cellular effects in HCAEC and CASMC cultures. While μF-CA and μF-CA/TvE microfibers affected CASMC viability, HCAEC culture was unaffected. However, cell proliferation was increased in the HCAEC culture. In contrast, the CASMC culture was inhibited when µF-CA/TvE was used, indicating a selective effect that could potentially mitigate muscle proliferation, a critical factor in atherosclerosis. Moreover, neither the CA nor the TvE-loaded systems affected platelet viability. However, both demonstrated the potential to reduce platelet activation and adhesion associated with CD40 release, inhibition of integrin αIIbβ3, and PECAM-1 expression. This indicates that they may serve as promising antithrombotic agents. The combined effects of CA and TvE may facilitate vascular healing and inhibit pathological processes, such as thrombosis, which is of particular importance in cardiovascular health. In order to ensure the reliability of the findings, further in-depth characterization is planned in future research. This includes advanced surface analysis techniques, such as Atomic Force Microscopy (AFM) and Fourier-transform infrared spectroscopy (FTIR), for chemical composition assessment to assess the nanostructural features of the coatings and their interactions with biological environments at the micro and nanoscale.

The findings collectively indicate that CA- and TvE-loaded PCL microfibers can modulate endothelial and smooth muscle cell behavior while promoting antiaggregatory effects on platelets. Future research will aim to elucidate the underlying mechanisms of these antithrombotic effects and further develop these systems for therapeutic applications in cardiovascular diseases.

## Figures and Tables

**Figure 1 ijms-26-04949-f001:**
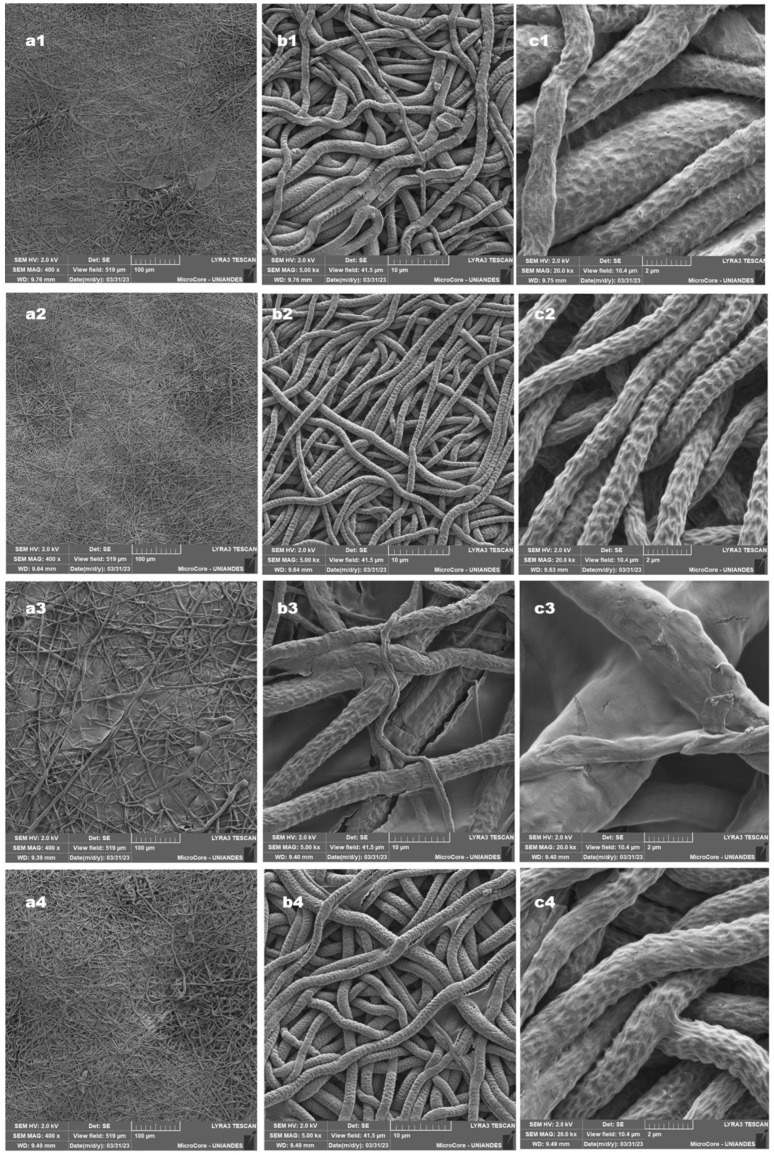
SEM images from microfiber systems. µF (**a1**–**c1**), µF-CA-1 (**a2**–**c2**), µF-TvE (**a3**–**c3**), and µF-CA-1/TvE (**a4**–**c4**).

**Figure 2 ijms-26-04949-f002:**
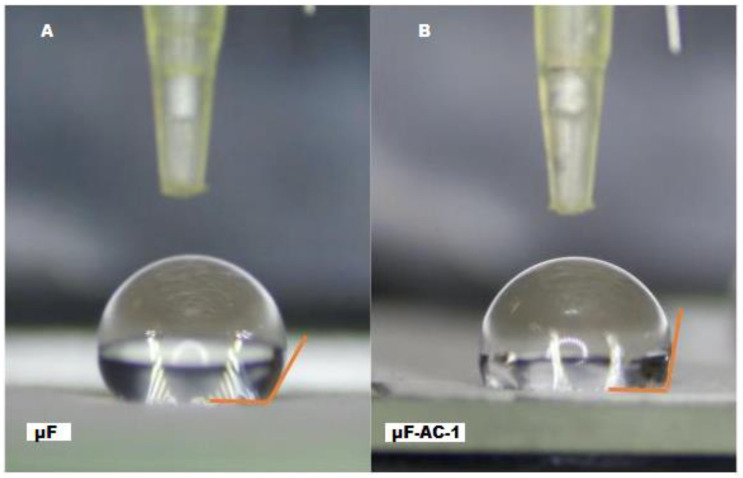
Measurement of contact angles. (**A**) A drop water on microfiber, (**B**) A drop water on microfiber AC-1.

**Figure 3 ijms-26-04949-f003:**
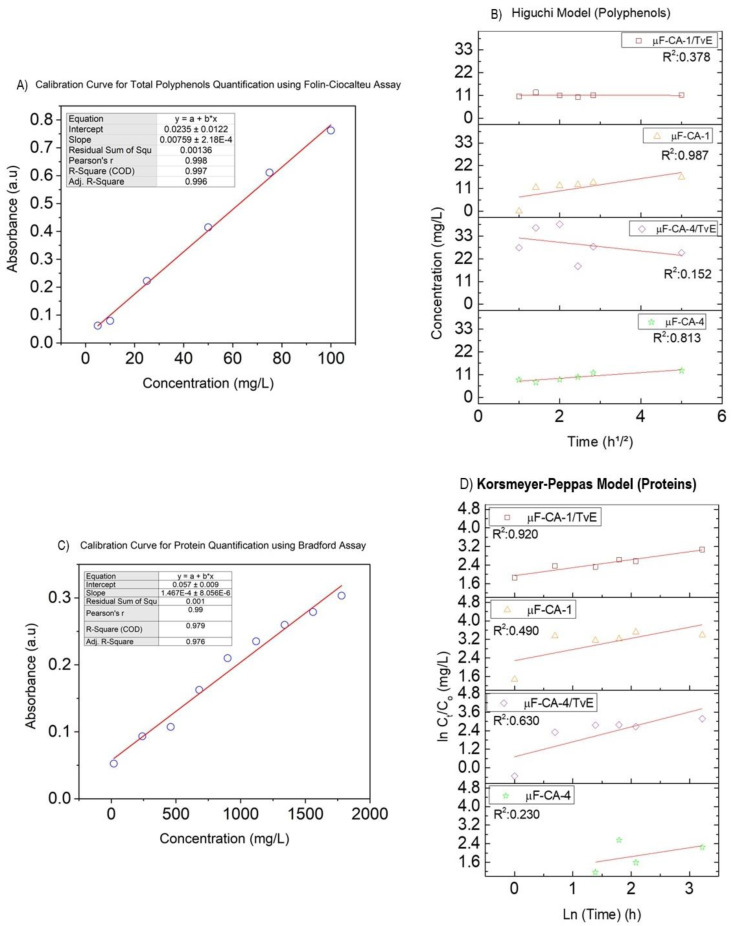
(**A**) Calibration curve for total polyphenol quantification using the Folin–Ciocalteu assay, (**B**) Calibration curve for total protein quantification using the Bradford assay, (**C**) Higuchi model for polyphenols, and (**D**) Korsmeyer–Peppas model for proteins.

**Figure 4 ijms-26-04949-f004:**
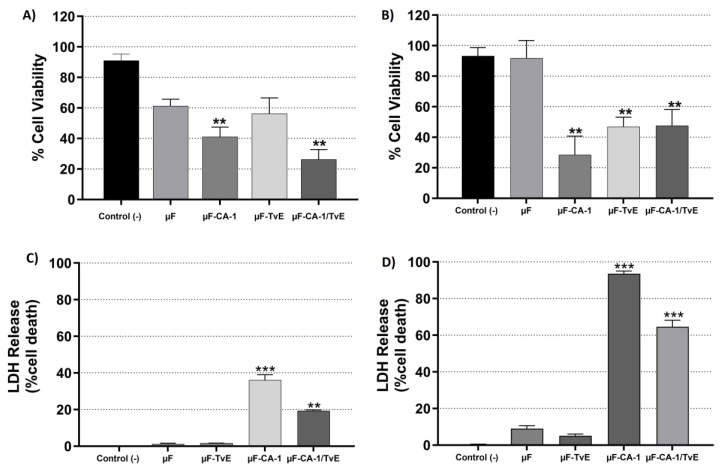
Cytotoxic activity and LDH release from CASMCs treated with 8% PCL microfibers and CA-1. Cell viability at (**A**) 24 h and (**B**) 48 h. LDH release at (**C**) 24 h and (**D**) 48 h. Bar represents mean percentage ± s.e.m. of three independent experiments in triplicate for each treatment. Asterisks indicates the difference compared with the control (unstimulated cells). ** (*p* < 0.01), *** (*p* < 0.001).

**Figure 5 ijms-26-04949-f005:**
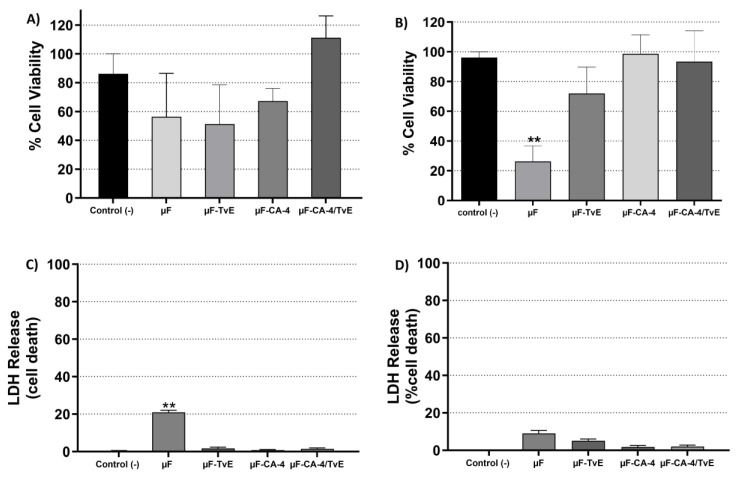
Cytotoxic activity and LDH release from CASMCs treated with 5% PCL microfibers and CA-4. Cell viability at (**A**) 24 h and (**B**) 48 h. LDH release at (**C**) 24 h and (**D**) 48 h. Bar represents mean percentage ± s.e.m. of three independent experiments in triplicate for each treatment. Asterisks indicates the difference compared with the control (unstimulated cells). ** (*p* < 0.01).

**Figure 6 ijms-26-04949-f006:**
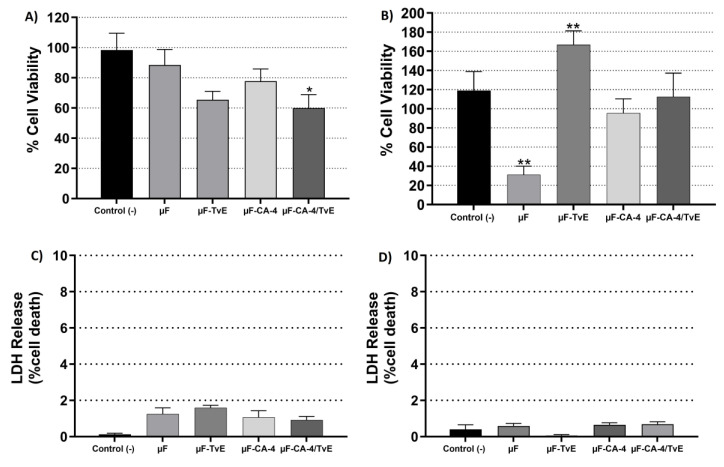
Cytotoxic activity and LDH release from HCAECs treated with 5% PCL microfibers and CA-4. Cell viability at (**A**) 24 h and (**B**) 48 h. LDH release at (**C**) 24 h and (**D**) 48 h. Bar represents mean percentage ± s.e.m. of three independent experiments in triplicate for each treatment. Asterisks indicates the difference compared with the control (unstimulated cells). * (*p* < 0.05), ** (*p* < 0.01).

**Figure 7 ijms-26-04949-f007:**
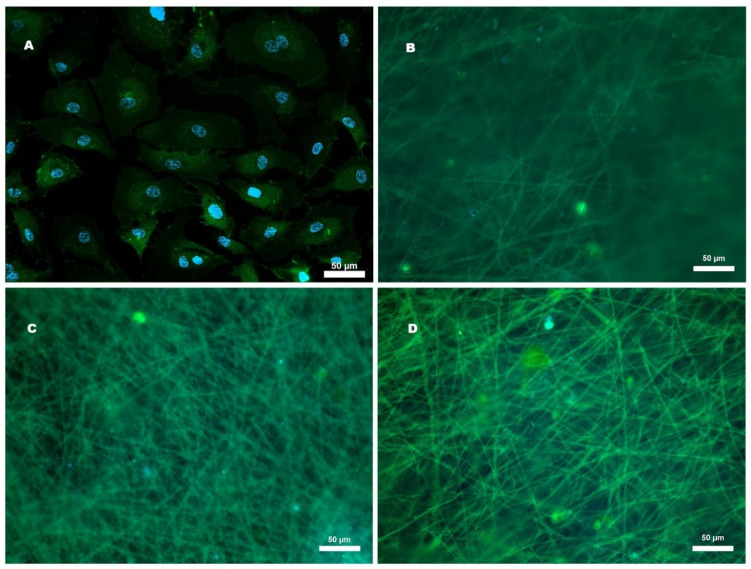
Expression of PECAM-1 (green) in HCAECs. Nuclei are stained with DAPI (blue). (**A**) Cells cultured without the delivery system. (**B**) Cells cultured on μF and (**C**) μF-CA-1. (**D**) Cells cultured on μF-CA-4. Cells were seeded and cultured for 7 d.

**Figure 8 ijms-26-04949-f008:**
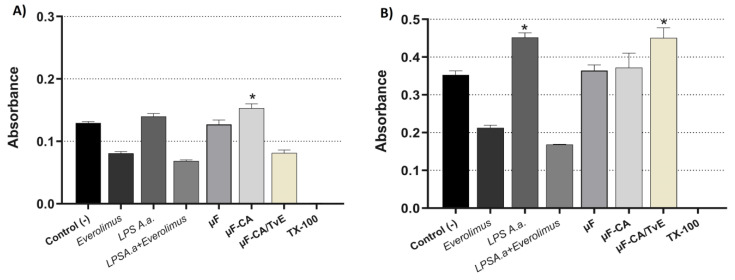
Proliferation of HCAECs treated with microfibers (μF) containing CA and TvE. After treatment for 24 h (**A**) and 48 h (**B**). (*) indicates the difference compared with the control (*p* < 0.05).

**Figure 9 ijms-26-04949-f009:**
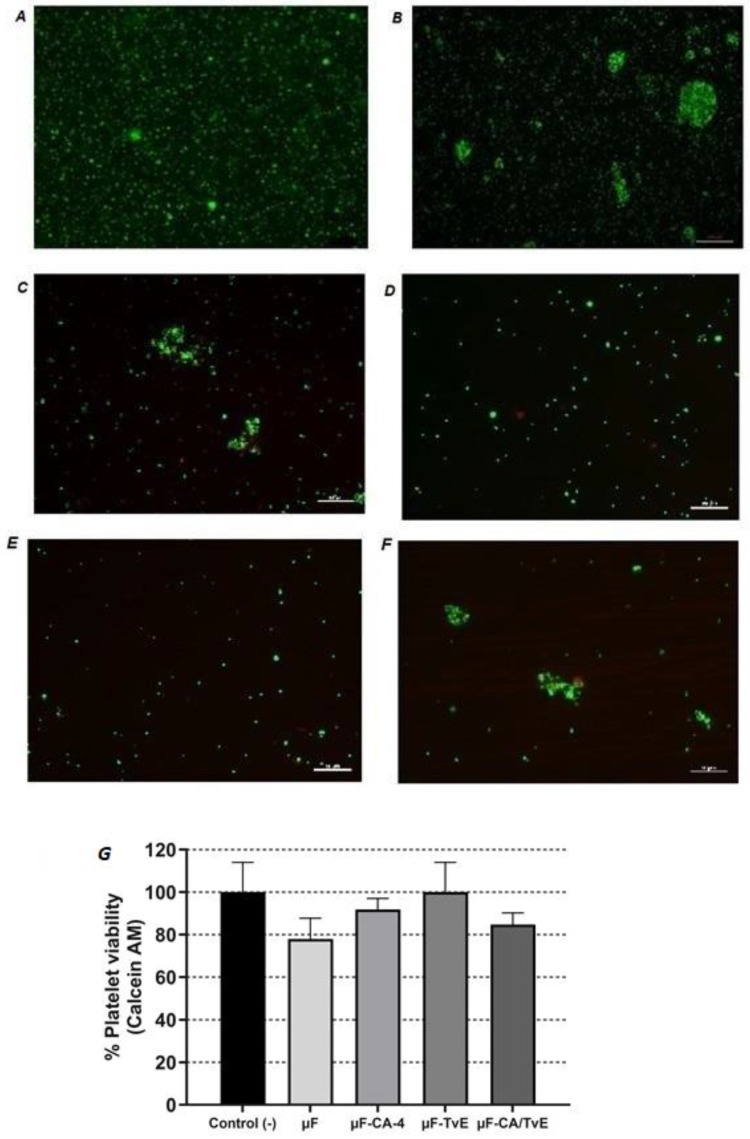
LIVE/DEAD staining to evaluate the viability of platelets stimulated with *P. gingivalis* LPS and treated with 5% PLC microfibers for 1 h. (**A**) Unstimulated platelets, (**B**) platelets stimulated with *P. gingivalis* LPS, (**C**) platelets stimulated with *P. gingivalis* LPS + μF, (**D**) platelets stimulated with *P. gingivalis* LPS + μF-CA, (**E**) platelets stimulated with *P. gingivalis* LPS + μF-TvE, and (**F**) platelets stimulated with *P. gingivalis* LPS + μF-CA/TvE. Images were photomicrographed with a fluorescence microscope at 100×. (**G**) Percentage of cell death compared with the control (unstimulated cells). Bar represents mean percentage ± s.e.m. from three independent experiments in triplicate for each treatment.

**Figure 10 ijms-26-04949-f010:**
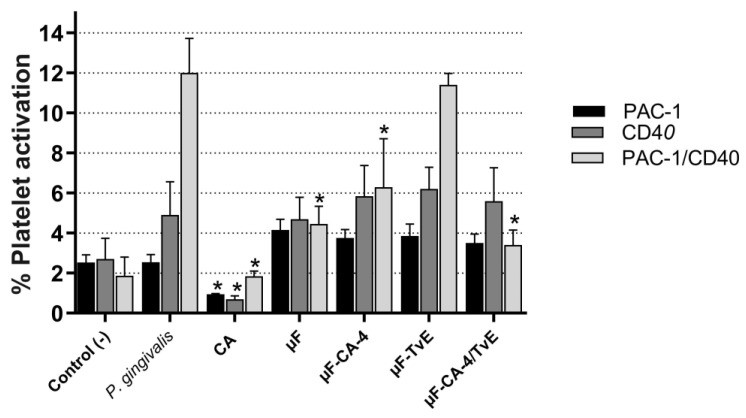
Platelets activated after stimulation with *P. gingivalis* LPS and treatment with 5% PCL loaded with CA or *T. versicolor* extract for 1 h. Bar represents the mean ± s.e.m. from three independent experiments in duplicate. (*) indicates differences compared with the effect of *P. gingivalis* (*p* < 0.05).

**Figure 11 ijms-26-04949-f011:**
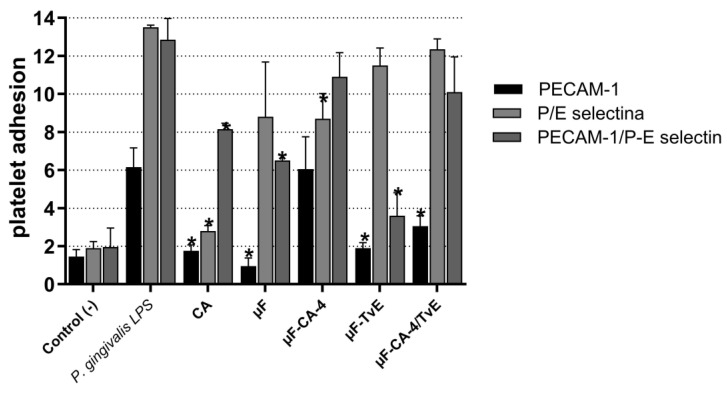
Percentage of PECAM-1 and P/E-selectin expression in platelets stimulated with *P. gingivalis* LPS and treated with PCL microfibers loaded with caffeic acid and/or *T. versicolor* for 1 h. Bar represents the average ± s.e.m. from three independent trials in duplicate. (*) indicates statistically significant difference compared with the effect of *P. gingivalis* (*p* < 0.05).

**Table 1 ijms-26-04949-t001:** Determination of fiber diameter and contact angle from microfiber systems.

System	Fiber Diameter Media ± S.D (µm)	D (min) (µm)	D (max) (µm)	Contact Angle
µF	2.50 ± 1.12	0.48	4.60	105.5° ± 3.25° (<90°)
µF-CA-1	1.96 ± 0.80	0.50	2.10	78.32° ± 2.84° (<90°)
µF-TvE	2.41 ± 1.53	0.95	4.21	not determined
µF-CA-1/TvE	2.03 ± 1.05	0.75	3.25	not determined

**Table 2 ijms-26-04949-t002:** Percentage of platelet activation molecules expressed by platelets infected with *P. gingivalis* LPS and treated with 5% polycaprolactone (PCL) microfibrils (μF). Human platelets were stimulated with *P. gingivalis* LPS (3.5 µg/mL) and treated with the active ingredient of caffeic acid (CA; 50 µM) or μF of PCL loaded with CA and/or *Trametes versicolor.* Results are presented as mean ± SEM. * Represents the percentage of platelet activation observed in three independent assays conducted in duplicate, compared with the effect of *P. gingivalis* (*p* ≤ 0.05).

Treatment	PAC-1 (%)	CD40 (%)	PAC-1/CD40 (%)
Control (−)	2.52 ± 0.025	2.70 ± 1.3	1.87 ± 1.0
*P. gingivalis* LPS	2.55 ± 1.5	4.90 ± 0.8	12 ± 0.4
*Caffeic acid*	0.95 ± 1.8 *	0.70 ± 2.2 *	1.85 ± 0.5 *
μF	4.15 ± 1.3	4.70 ± 2.9	4.45 ± 0.9
μF-CA-4	3.75 ± 0.8	5.85 ± 2.4	6.30 ± 1.4 *
μF-TvE	3.85 ±2.2	6.2 ± 3.8	11.4 ± 2.2
μF-CA-4/TvE	3.5 ± 1.6	5.6 ± 0.5	3.4 ± 2.0 *

**Table 3 ijms-26-04949-t003:** Expression (%) of platelet adhesion molecules infected with *P. gingivalis* LPS and treated with 5% PCL microfibers. Additionally, the effects of caffeic acid (CA; 50 µM) and PCL microfibers loaded with CA and/or *T. versicolor* on human platelets stimulated with *P. gingivalis* LPS (3.5 µg/mL) are shown. Results are presented as mean ± SEM. * Represents the percentage of platelet adhesion compared with the effect of *P. gingivalis* (*p* ≤ 0.05).

Treatment	PECAM-1 (%)	P/E Selectina (%)	PECAM-1/P-E Selectina (%)
Control (+)	1.45 ± 0.3	1.90 ± 0.15	1.95 ± 0.55
*P. gingivalis* LPS	6.15 ± 2.4	13.5 ± 2.3	12.8 ± 2.0
*Caffeic acid*	1.75 ± 0.15 *	2.80 ± 2.1 *	8.15 ± 2.5 *
μF	0.95 ± 0.3 *	8.8 ± 3.4	6.5 ± 1.1 *
μF-CA-4	6.0 ± 0.25	8.75 ± 0.25 *	10.9 ± 1.4
μF-TvE	1.70 ± 0.3 *	11.5 ± 4.4	3.60 ± 4.4 *
μF-CA-4/TvE	3.0 ± 2.2 *	12.3 ± 0.9	10.0 ± 0.65

## Data Availability

The data presented in this study are available on request from the corresponding authors.

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
