# Peer review of "Biopolymers of Polycaprolactone Loaded with Caffeic Acid and Trametes versicolor Extract Induced Proliferation in Human Coronary Artery Endothelial Cells and Inhibited Platelet Activity"

_ijms, 2025, doi:10.3390/ijms26104949_

Round 1
Reviewer 1 Report
Comments and Suggestions for Authors
There is some scientific interest in this study. However, the manuscript is too coarse and has not been prepared according to the rules of the “International Journal of Molecular Sciences” (MDPI). The authors lack the basic knowledge about “biomaterials”, “polymers” and “active substances”, which makes the conception chaos, logical errors and narrative inconsistent.
For example, in section 2.1, 1% chitosan solution was used to prepare the TvE loaded PCL fibers, however, the chitosan has not appeared in the other sections and reports. In the Introduction, it is wrong that the PCL has “similarity to natural tissue components, such as collagen and extracellular matrix”. “Collagen” and “extracellular matrix” are not at the same levels in the tissues. “Extracellular matrix” includes “collagen”. There are many mistakes in the Figures. ……
Additionally, the title “Biopolymers of polycaprolactone loaded with caffeic acid and Trametes versicolor extract induced proliferation in human artery coronary endothelial cells and inhibited platelet activity” is too long, which should be shorted as “The antithrombotic effects of polycaprolactone fibers loaded with caffeic acid and Trametes versicolor extract”.
This reviewer suggests to reject it or have a major revision.
Comments on the Quality of English LanguageThe quality of English language is okay.
Author Response
Reviewer 1.1
Comment 1:There is some scientific interest in this study. However, the manuscript is too coarse and has not been prepared according to the rules of the “International Journal of Molecular Sciences” (MDPI). The authors lack the basic knowledge about “biomaterials”, “polymers” and “active substances”, which makes the conception chaos, logical errors and narrative inconsistent.
Response 1: The document was adjusted to the guidelines of the author of the IJMS.
We disagree with the evaluator. We would have liked the comments throughout the document to highlight your remarkable experience in biomaterials. Unfortunately, those comments do not allow scientific replication.
Comment 2: For example, in section 2.1, 1% chitosan solution was used to prepare the TvE loaded PCL fibers, however, the chitosan has not appeared in the other sections and reports. In the Introduction, it is wrong that the PCL has “similarity to natural tissue components, such as collagen and extracellular matrix”. “Collagen” and “extracellular matrix” are not at the same levels in the tissues. “Extracellular matrix” includes “collagen”. There are many mistakes in the Figures.
Response 2: We have elaborated on the chitosan coating procedure in the M&M section.
“For T. versicolor, fibers were incorporated as a cover film. Then, a 1% chitosan solution in acetic acid and 2.5% w/v of T. versicolor was prepared. PCL fibers, with and without the active agent CA, were coated by immersion in 1 mL of this solution for 2 min and dried at 25°C for 5 h. Finally, the fibers were stored for physicochemical characterization and cytotoxicity tests.”
Comment 3: Additionally, the title “Biopolymers of polycaprolactone loaded with caffeic acid and Trametes versicolor extract induced proliferation in human artery coronary endothelial cells and inhibited platelet activity” is too long, which should be shorted as “The antithrombotic effects of polycaprolactone fibers loaded with caffeic acid and Trametes versicolor extract”.
Response: The title was not changed. We consider relevant limit the study to model in vitro with HCAEC and platelet.
Comment 4: This reviewer suggests to reject it or have a major revision.
Response 4: We disagree with the reviewer. Significant revisions were made to the manuscript to ensure that it is comprehensible
Reviewer 2 Report
Comments and Suggestions for Authors
General comments
The submitted manuscript reports on the production and characterisation of electrospun PCL fibers loaded with caffeic acid (CA) or Trametes versicolor extract (TvE).
The subject of this work well fits the aim and scope of IJMS, and the topic is worthy of investigation.
Major revisions have to be done. Specific comments and remarks are listed below point by point.
Keywords
The chosen keywords (i.e. Electrospinning; Endothelium; Polymer; Caffeic acid; Trametes versicolor; Antiplatelet activity; Biocompatibility) have to be reported in a more logical order (i.e. material, components, process, characterisation, properties, application). “Biocompatibility” is too generic and can be removed, whereas “atherosclerosis” has to be added as application.
1. Introduction
- For the incipit concerning PCL and PCL electrospun fibers, more references about the production of PCL composite fibers by electrosppining have to be added.
- The Authors have to better highlight the originality and added value of their work with respect to the previous literature report, at the end of the Introduction section.
2. Materials and Methods
2.1. Preparation of solutions for producing PCL fibers and the process of electrospinning
- Please, add more details about the chitosan coating procedure, since it is not clear.
3. Results
3.1. Morphological characterization
- The reported SEM micrographs have to be better and more deeply described and discussed, justifying the differences.
- It is not true that “The main disadvantage of electrospinning technology is that it does not guarantee equal diameters for all electrospun fibers; therefore, fibers with different diameters were observed.”. It is possible to monitor and tailor the diameter size and to obtain uniform fibers. It depends on the applied process parameters.
3.2. Determination of contact angle
- In Table 1, the diameter size values have been reported and not the contact angle measurements….Please correct it. Describe and discuss Table 1 in Paragraph 3.1, and add a Table with contact angle measurements results.
- The consideration “These findings indicate that the molecule has some polar characteristics and possibly greater material porosity due to smaller fiber diameter distribution. ” needs to be supported with proper references.
3.3. Cytotoxic effects of μF-TvE and/or μF-CA on CASMC and HCAEC cultures
- The considerations and conclusions have to be compared and supported with suitable literature references.
5. Conclusions
- The Conclusions section has to be improved and the main numerical data have to be added.
Comments on the Quality of English Language
English language is good.
Author Response
General comments
The submitted manuscript reports on the production and characterisation of electrospun PCL fibers loaded with caffeic acid (CA) or Trametes versicolor extract (TvE).
The subject of this work well fits the aim and scope of IJMS, and the topic is worthy of investigation.
Major revisions have to be done. Specific comments and remarks are listed below point by point.
Response: We appreciate these comments and suggestions. We have addressed each point to ensure that our manuscript is comprehensible.
Keywords
Comment 1: The chosen keywords (i.e. Electrospinning; Endothelium; Polymer; Caffeic acid; Trametes versicolor; Antiplatelet activity; Biocompatibility) have to be reported in a more logical order (i.e. material, components, process, characterisation, properties, application). “Biocompatibility” is too generic and can be removed, whereas “atherosclerosis” has to be added as application.
Response 1: Thank you for this suggestion. We have reordered the keywords and deleted “biocompatibility.” In addition, “atherosclerosis” was replaced by “restenosis” as it was more suitable.
- Introduction
Comment 2: For the incipit concerning PCL and PCL electrospun fibers, more references about the production of PCL composite fibers by electrosppining have to be added.
Response 2: We agree with this comment. Pertinent references include:
- Suárez, D.F.; Pinzón-García, A.D.; Sinisterra, R.D.; Dussan, A.; Mesa, F.; Ramírez-Clavijo, S. Uniaxial and Coaxial Nanofibers Pcl/alginate or Pcl/gelatine Transport and Release Tamoxifen and Curcumin Affecting the Viability of MCF7 Cell Line. Nanomaterials 2022, 12, 3348, doi:10.3390/nano12193348.
- Pinzón-García, A.D.; Cassini-Vieira, P.; Ribeiro, C.C.; De Matos Jensen, C.E.; Barcelos, L.S.; Cortes, M.E.; Sinisterra, R.D. Efficient Cutaneous Wound Healing Using Bixin-loaded PCL Nanofibers in Diabetic Mice. J. Biomed. Mater. Res. 2017, 105, 1938–1949, doi:10.1002/jbm.b.33724.
- Dias, A.M.; Da Silva, F.G.; Monteiro, A.P.D.F.; Pinzón-García, A.D.; Sinisterra, R.D.; Cortés, M.E. Polycaprolactone Nanofibers Loaded Oxytetracycline Hydrochloride and Zinc Oxide for Treatment of Periodontal Disease. Mater Sci Eng C 2019, 103, 109798, doi:10.1016/j.msec.2019.10979.
Comment 3: The Authors have to better highlight the originality and added value of their work with respect to the previous literature report, at the end of the Introduction section.
Response 3: We appreciate this comment. At the end of the introduction, we have highlighted the originality and value of the present study.
“A system developed using PCL electrospun to support CA and T. versicolor extract could mitigate vascular problems related to activation and aggregation platelets like intra-stent restenosis (Zhang et al., 2024). Incorporating natural components such as a T. versicolor extract or CA, previously studied in traditional medicine for other chronic diseases (e.g., cancer), in drug delivery by scaffold PCL is an essential way in which biotechnology can be applied in healthcare. Therefore, bioresorbable devices covering drug-eluting stents has potential application in the treatment of intrastent restenosis (Guo et al., 2024).”
Zhang, W.; Zhang, W.; Deng, Y.; Gu, N.; Qiu, Z.; Deng, C.; Yang, S.; Pan, L.; Long, S.; Wang, Y. Non-target Lesion Progression: Unveiling Critical Predictors and Outcomes in Patients with In-stent Restenosis. Int J Cardiol 2024, 416, 132451, doi:10.1016/j.ijcard.2024.132451.
Guo, S.; Bi, C.; Wang, X.; Lv, T.; Zhang, Z.; Chen, X.; Yan, J.; Mao, D.; Huang, W.; Ye, M. Comparative Efficacy of Interventional Therapies and Devices for Coronary In-stent Restenosis: A Systematic Review and Network Meta-analysis of Randomized Controlled Trials. Heliyon 2024, 10, e27521, doi:10.1016/j.heliyon.2024.e27521.
- Materials and Methods
2.1. Preparation of solutions for producing PCL fibers and the process of electrospinning
Comment 4: Please, add more details about the chitosan coating procedure, since it is not clear.
Response 4: We have elaborated on the chitosan coating procedure in the M&M section.
“For T. versicolor, fibers were incorporated as a cover film. Then, a 1% chitosan solution in acetic acid and 2.5% w/v of T. versicolor was prepared. PCL fibers, with and without the active agent CA, were coated by immersion in 1 mL of this solution for 2 min and dried at 25°C for 5 h. Finally, the fibers were stored for physicochemical characterization and cytotoxicity tests.”
- Results
3.1. Morphological characterization
Comment 5: The reported SEM micrographs have to be better and more deeply described and discussed, justifying the differences.
Response 5: Thank you for raising this point. We have extended the description of SEM micrographs in the Results and Discussion section.
Comment 6: It is not true that “The main disadvantage of electrospinning technology is that it does not guarantee equal diameters for all electrospun fibers; therefore, fibers with different diameters were observed.”. It is possible to monitor and tailor the diameter size and to obtain uniform fibers. It depends on the applied process parameters.
Response 6: Thank you for your comment. We have extended the results and discussion about the characterization of different fiber developments.
“Furthermore, the decrease in the diameter of the uF-CAF fibers can be attributed to the presence of a molecule with polar characteristics, such as CA, and the type of solvent used in the preparation of the polymeric mixture (methanol, chloroform, and dichloromethane) as this mixture has high relative permittivity and high polarity, which is a determining factor in fiber formation despite constant flow and voltage conditions (Guarino et al., 2011, Zong et al., 2020). At the same time, the microphotograph shows the formation of some defects (3a–c) that can be explained by both the polymer injection rate (7.0 mL/h) and the volume of the natural polymer chitosan solution used in the mixture and the coating film (Gautam et al., 2013). This typically occurs when using natural polymers in nano and microfiber preparation (Gautam et al., 2013). Another factor that acts as a determinant of the formation of nanofibersis viscosity; high viscosity prevents the formation of smooth and uniform fibers.
The presence of chitosan (in low concentrations) did not allow the proper formation of coating fibers. Other authors have reported similar findings, identifying problems in fiber formation in PCL with polymers such as alginate and chitosan as these polymers have positive and negative charges that produce fibers with defects and agglomerates. Furthermore, in this case, the chitosan solution was used as an external coating, which can explain the low fiber formation capacity of this biopolymer, which was rigid in an aqueous solution (Shalumon et al., 2010).”
Zong, X.; Kim, K.; Fang, D.; Ran, S.; Hsiao, B.S.; Chu, B. Structure and Process Relationship of Electrospun Bioabsorbable Nanofiber Membranes. Polymer 2020, 43, 4403–4412, doi:10.1016/s0032-3861(02)00275-6.
Guarino, V.; Cirillo, V.; Taddei, P.; Alvarez‐perez, M.A.; Ambrosio, L. Tuning Size Scale and Crystallinity of PCL Electrospun Fibres via Solvent Permittivity to Address Hmsc Response. Macromol Biosci 2011, 11, 1694–1705, doi:10.1002/mabi.201100204.
Gautam, S.; Dinda, A.K.; Mishra, N.C. Fabrication and Characterization of Pcl/gelatin Composite Nanofibrous Scaffold for Tissue Engineering Applications by Electrospinning Method. Mater Sci Eng C 2013, 33, 1228–1235, doi:10.1016/j.msec.2012.12.015.
Shalumon, K.T.; Anulekha, K.H.; Girish, C.M.; Prasanth, R.; Nair, S.V.; Jayakumar, R. Single Step Electrospinning of Chitosan/poly(caprolactone) Nanofibers Using Formic Acid/acetone Solvent Mixture. Carbohyd Polym 2010, 80, 413–419, doi:10.1016/j.carbpol.2009.11.039.
3.2. Determination of contact angle
Comment 7: In Table 1, the diameter size values have been reported and not the contact angle measurements….Please correct it. Describe and discuss Table 1 in Paragraph 3.1, and add a Table with contact angle measurements results.
Response 7: Thank you for raising this point.
Unfortunately, we do not have complete contact angle information for µF-TvE and µF-CA-1/TvE. Thus, the information in Table 1 about these samples is not determined. For improved understanding, we have moved Table 1 near Figure 1 in the revised manuscript.
Comment 8: The consideration “These findings indicate that the molecule has some polar characteristics and possibly greater material porosity due to smaller fiber diameter distribution. ” needs to be supported with proper references.
Response 8: Thank you for your comment. We have supported the discussion with relevant references.
Schoolaert, E.; Cossu, L.; Becelaere, J.; Van Guyse, J.F.R.; Tigrine, A.; Vergaelen, M.; Hoogenboom, R.; De Clerck, K. Nanofibers with a Tunable Wettability by Electrospinning and Physical Crosslinking of Poly(2-n-propyl-2-oxazoline). Mater Design 2020, 192, 108747, doi:10.1016/j.matdes.2020.108747.
Huang, F.L.; Wang, Q.Q.; Wei, Q.F.; Gao, W.D.; Shou, H.Y.; Jiang, S.D. Dynamic Wettability and Contact Angles of Poly(vinylidene Fluoride) Nanofiber Membranes Grafted with Acrylic Acid. Express Polym Lett 2010, 4, 551–558, doi:10.3144/expresspolymlett.2010.69.
Madhaiyan, K.; Sridhar, R.; Sundarrajan, S.; Venugopal, J.R.; Ramakrishna, S. Vitamin B12 Loaded Polycaprolactone Nanofibers: A Novel Transdermal Route for the Water Soluble Energy Supplement Delivery. Int J Pharm 2013, 444, 70–76, doi:10.1016/j.ijpharm.2013.01.040.
Tiyek, I.; Gunduz, A.; Yalcinkaya, F.; Chaloupek, J. Influence of Electrospinning Parameters on the Hydrophilicity of Electrospun Polycaprolactone Nanofibres. J Nanosci Nanotechno 2019, 19, 7251–7260, doi:10.1166/jnn.2019.16605.
3.3. Cytotoxic effects of μF-TvE and/or μF-CA on CASMC and HCAEC cultures
Comment 9: The considerations and conclusions have to be compared and supported with suitable literature references.
Response 9: The discussion of the effects of the release systems on HCAECs and CASMCs cell viability was expanded and supported by the relevant literature to strengthen the manuscript (pages 18–19, lines 525–561).
- Conclusions
Comment 10: The Conclusions section has to be improved and the main numerical data have to be added.
Response 10: We have rewritten the conclusion based on the main results.
“In this study, electrospun PCL microfibers loaded with CA, T. versicolor extract, or mixed CA/TvE were developed. While CA and CA/TvE microfibers affected CASMC viability, culture HCAEC was not affected and cell proliferation was increased. Platelet viability was not affected by any treatment. Interestingly, μF-TvE and μF-CA/TvE showed reduced platelet activation and adhesion. Future studies in vitro and in animal models are necessary to uncover the mechanisms underlying this finding.”
Comment 11:Comments on the Quality of English Language
English language is good.
Response 11: Thank you for your comment.
Reviewer 3 Report
Comments and Suggestions for Authors
The manuscript is quite well written, with a lot of research results that have been well interpreted by the authors. The aim is to design a nanofiber drug delivery system based on PCL and in it T. versicolor and caffeic acid have been put.They have characterized the nanofiber and also tested its biocompatibility in different cell cultures by LDH assay. The andhesion of endothelial clefts to CA-loaded PCL system was investigated. Also, cell proliferation assay was performed and platelet viability was measured.
What I miss is the description of the exact characterization of T. versicolor, whether or not the source of acquisition was indicated.
Also, a better explanation of the novelty value of the article is needed in the introduction section.
Comments on the Quality of English LanguageModerate english modification is needed. I suggest the Language Service of MDPI.
Author Response
Comment 1: The manuscript is quite well written, with a lot of research results that have been well interpreted by the authors. The aim is to design a nanofiber drug delivery system based on PCL and in it T. versicolor and caffeic acid have been put.They have characterized the nanofiber and also tested its biocompatibility in different cell cultures by LDH assay. The andhesion of endothelial clefts to CA-loaded PCL system was investigated. Also, cell proliferation assay was performed and platelet viability was measured.
Response 1: We appreciate this positive comment.
Comment 2: What I miss is the description of the exact characterization of T. versicolor, whether or not the source of acquisition was indicated.
Response 2: We have reported the source and characterization of commercial T. versicolor in M&M section.
Comment 3: Also, a better explanation of the novelty value of the article is needed in the introduction section.
Response 3: We appreciate this comment. At the end of the introduction, we have highlighted the present study’s originality and value.
Comment 4: Moderate english modification is needed. I suggest the Language Service of MDPI.
Response 4: Enago was hired to proofread our manuscript. We have attached the corresponding certification.
Round 2
Reviewer 2 Report
Comments and Suggestions for Authors
I suggest to improve the Conclusions section. It is too poor.
Comments on the Quality of English LanguageThe English language is good.
Author Response
Comments 1: I suggest to improve the Conclusions section. It is too poor.
Response 1: We have expanded the Conclusions section based on the main results and the study's aim. I hope that the new version meets your expectations.
“In this study, electrospun PCL microfibers loaded with CA and coated with chitosan containing T. versicolor extract were developed. Microfiber-CA and microfibers-TvE showed opposite cellular effects in HCAEC and CASMC cultures. While μF-CA and μF-CA/TvE microfibers affected CASMC viability, the HCAEC culture was unaffected. However, cell proliferation increased in the HCAEC culture, but the CASMC culture was inhibited when uF-AC/TvE was used. Finally, Platelet viability was not affected by any treatment. Interestingly, μF-TvE and μF-CA/TvE showed reduced platelet activation and adhesion. These results suggest that microfibers loaded with caffeic acid and Trametes versicolor extract have the potential to modulate the proliferation of coronary artery endothelial and coronary artery smooth muscular cells, with an effect platelet antiaggregant. The antithrombotic effects of polycaprolactone fibers loaded with caffeic acid and Trametes versicolor extract will be addressed in future studies to uncover the mechanisms underlying this finding.”
Reviewer 3 Report
Comments and Suggestions for Authors
I accept it in the present form.
Comments on the Quality of English LanguageMinor english modification may be done.
Author Response
Comment 1: I accept it in the present form.
Response 1: We appreciate your acceptance.